# EPA is More Effective than DHA to Improve Depression-Like Behavior, Glia Cell Dysfunction and Hippcampal Apoptosis Signaling in a Chronic Stress-Induced Rat Model of Depression

**DOI:** 10.3390/ijms21051769

**Published:** 2020-03-05

**Authors:** Zhilan Peng, Cai Zhang, Ling Yan, Yongping Zhang, Zhiyou Yang, Jiajia Wang, Cai Song

**Affiliations:** 1Research Institute for Marine Drugs and Nutrition, College of Food Science and Technology, Guangdong Ocean University, Zhanjiang 524088, Chinazhangcai910206@163.com (C.Z.); ling.yan@ut.ee (L.Y.); zhangyp2015@163.com (Y.Z.); yang_zhiyou@sina.com (Z.Y.);; 2Shenzhen Institutes of Guangdong Ocean University, Shenzhen 518120, China; 3Guangdong Provincial Key Laboratory of Aquatic Product Processing and Safety, College of Food Science and Technology, Guangdong Ocean University, Zhanjiang 524088, China

**Keywords:** EPA, DHA, depression, neuroinflammation, neurotrophins, microglia, astrocytes, BDNF

## Abstract

Clinical evidence indicated that eicosapentaenoic acid (EPA) was more effective than docosahexaenoic acid (DHA) in depression treatment. However, possible mechanisms remain unclear. Here, a chronic unpredictable mild stress (CUMS)-induced model of depression was used to compare EPA and DHA anti-depressant effects. After EPA or DHA feeding, depression-like behavior, brain n-3/n-6 PUFAs profile, serum corticosterone and cholesterol concentration, hippocampal neurotransmitters, microglial and astrocyte related function, as well as neuronal apoptosis and survival signaling pathways were studied. EPA was more effective than DHA to ameliorate CUMS-induced body weight loss, and depression-like behaviors, such as increasing sucrose preference, shortening immobility time and increasing locomotor activity. CUMS-induced corticosterone elevation was reversed by bother fatty acids, while increased cholesterol was only reduced by EPA supplement. Lower hippocampal noradrenaline and 5-hydroxytryptamine concentrations in CUMS rats were also reversed by both EPA and DHA supplement. However, even though CUMS-induced microglial activation and associated increased IL-1β were inhibited by both EPA and DHA supplement, increased IL-6 and TNF-α levels were only reduced by EPA. Compared to DHA, EPA could improve CUMS-induced suppressive astrocyte biomarkers and associated BDNF-TrkB signaling. Moreover, EPA was more effective than DHA to attenuate CUMS-induced higher hippocampal NGF, GDNF, NF-κB, p38, p75, and bax expressions, but reversed bcl-2 reduction. This study for the first time revealed the mechanisms by which EPA was more powerful than DHA in anti-inflammation, normalizing astrocyte and neurotrophin function and regulating NF-κB, p38 and apoptosis signaling. These findings reveal the different mechanisms of EPA and DHA in clinical depression treatment.

## 1. Introduction

Neuroinflammation and neurotrophin dysfunction play crucial roles in the etiology of depression [1]. Increased plasma levels of proinflammatory cytokines, such as interleukin (IL)-1β, IL-6 and tumor necrosis factor (TNF)-α, while decreased brain-derived neurotrophic factor (BDNF) have been reported in depressed patients [2,3].

As a trigger of depression, prolonged exposure to life stressful events can induce depression. Thus, chronic unpredictable mild stress (CUMS)-induced rodent model has been popularly used to study depression [4] and various treatment [5,6]. Continuous stimulation of the hypothalamic-pituitary adrenal (HPA) axis to release glucocorticoids during stress exposure can activate microglia to produce proinflammatory cytokines [7], which decrease serotonin (5-HT), dopamine (DA) and noradrenaline (NE) levels in the limbic system by activating inflammatory signaling pathways of indoleamine-2,3-dioxygenase (IDO) and mitogen-activated protein kinase (MAPK) [2,8,9]. Furthermore, glia-produced proinflammatory cytokines, as well as reactive oxygen species (ROS)/reactive nitrogen species (RNS) that arises from stress stimuli, may activate NF-κB and p38 mitogen-activated protein kinase (MAPK) pathways to trigger neuronal apoptosis through upregulating the bcl-2 family of apoptosis proteins, and caspases expression [10,11]. The changes induced by activated microglial cells may suppress the secretion of neurotrophins from astrocytes [12]. Among all neurotrophic factors, BDNF is the best-studied in depression, which exerts its effects through two specific receptors, tyrosine kinase receptor (Trk)B and p75. The activation of TrkB facilitates neuronal survival, differentiation and synaptogenesis [13,14], whereas p75 triggers apoptosis [15,16]. Therefore, anti-inflammatory/modulating neurotrophins may provide new strategy for the future prevention and treatment in depression.

Recent epidemiological investigations have reported that higher dietary intake of omega (n)-3 polyunsaturated essential fatty acids (PUFAs) are associated with decreased risk of depressive disorders [17,18]. Patients with lower n-3 PUFAs levels and higher serum n-6/n-3 PUFAs ratio are associated with greater depressive symptoms [19,20]. Previously, we have reported that n-3 PUFAs eicosapentaenoic acid (EPA) can reduce inflammation and up-regulate neurotrophin expression [21,22,23,24], thereby improving depression-like behaviors in olfactory bulbectomized rats, a valid depression model via anti-inflammation and the upregulation of nerve growth factor (NGF) [25]. As important components of cell membrane, both n-3 PUFAs EPA and docosahexaenoic acid (DHA) are potent activators of peroxisome proliferator-activated receptors (PPAR), which can inhibit inflammation [26]. Although EPA is far less abundant in the brain than DHA, it is highly neuroactive, as a meta-analysis reported that EPA may be more effective than DHA in the treatment of depression [27]. In addition, a randomized, controlled trial showed that EPA but not DHA pretreatment could significantly decrease IFN-α-induced depression-like changes [28]. Even though EPA and DHA showed different efficacies and effects in clinical treatment of depression, neuro-immunological and neurotrophic mechanisms by which EPA and DHA differently modulate microglial and astrocyte activity and function, brain PUFA profile, stress hormone, neurotransmitters, eventually affect depression-like behaviors, remain unknown.

Our previous studies have demonstrated that behavioral, glucocorticoid, neuroinflammatory, neurotrophin and neurotransmitter changes induced by CUMS are similar to those observed in depressed patients [6]. Thus, by using CUMS-induced depression model of rats, the present study aimed to compare and evaluate the different mechanism of EPA and DHA in the treatment of depression. Behavioral tests including the sucrose preference test (SPT), open field test (OFT), and forced swimming test (FST), were applied to evaluate the antidepressant effects of EPA or DHA. Brain n3/n6 PUFAs profile, serum corticosterone and cholesterol concentration, hippocampal monoamine neurotransmitter levels, hippocampal inflammatory cytokine concentrations, and expressions of glial marker CD11b, astrocyte marker glial fibrillary acidic protein (GFAP), neurotrophins, including glial cell-derived neurotrophic factor (GDNF), nerve growth factor (NGF) and BDNF, BDNF receptors TrkB and p75, NF-κB and p38 MAPK pathways, as well as neuron apoptosis (bax) and anti-apoptosis (bcl-2) signaling in the hippocampus were studied.

## 2. Results

### 2.1. Behavioral Measurements

Depressive-like behaviors were assessed by the sucrose preference test (SPT), open field test (OFT) and forced swimming test (FST), which are shown in Figure 1. Compared to control group, rats exposed to CUMS developed decreased sucrose consumption in the SPT (*p* < 0.01), increased immobility time in the FST (*p* < 0.01), as well as less numbers of crossing (*p* < 0.01) and rearing (*p* < 0.01) in the OPT. However, EPA exerted better effects than DHA to improve CUMS-induced depression-like behaviors changes.

In the SPT, decreased sucrose consumption induced by CUMS exposure (*p* < 0.01), was reversed by supplementation of EPA (*p* < 0.01) or DHA (*p* < 0.01). However, EPA feeding was more effective than DHA in the improvement of the anhedonia behavior, with a higher sucrose consumption. (*p* < 0.01) (Figure 1A).

In the FST, although longer immobility time induced by CUMS exposure (*p* < 0.01) was attenuated by supplementation of EPA (*p* < 0.01) or DHA (*p* < 0.05), compared with DHA feeding, EPA consumption was more effective to reduce immobility time in SPT (*p* < 0.05) (Figure 1B).

In the OFT, compared to control group, numbers of crossing and rearing in the open field arena were significantly decreased in rats subjected to CUMS (*p* < 0.01) (Figure 1C,D), which could only be improved by EPA, and no effect shown by DHA supplementation (Figure 1C,D).

### 2.2. n-3 and n-6 PUFAs Profiles in the Brain

The n-3 and n-6 PUFA profile in the brain were evaluated by gas chromatography (GC). As shown in Table 1, increased arachidonic acid (AA) level (*p* < 0.05), but decreased levels of docosapentaenoic acid (DPA) (*p* < 0.05), EPA (*p* < 0.05) and DHA (*p* < 0.01) were found in rats exposed to CUMS. On the one hand, increased AA content was attenuated by feeding rats with EPA or DHA diets (*p* < 0.05 respectively), while decreased EPA and DPA levels were reversed by EPA consumption but not by DHA (*p* < 0.05 respectively). On the other hand, the lower concentration of DHA caused by CUMS could only be improved by the intake of DHA but not by EPA (*p* < 0.05 respectively) (Table 1).

### 2.3. Changes of Serum Levels of Total Cholesterol (T-CHO) and Corticosterone

As shown by Figure 2, the contents of total cholesterol and corticosterone in the serum were significantly increased after CUMS administration (*p* < 0.01). Increased total cholesterol level could only be inhibited by EPA supplementation (*p* < 0.05) (Figure 2A), whereas increased serum corticosterone content could be attenuated by EPA or DHA supplementation *p* < 0.05 respectively) (Figure 2B).

### 2.4. Changes in the Concentration of Neurotransmitters and Metabolites in the Hippocampus

The data from high performance liquid chromatography (HPLC) analyses demonstrated that concentrations of NE, 5-HT, and the ratio of NE/MHPG in the hippocampus were decreased after CUMS exposure (*p* < 0.05 respectively), whereas the level of MHPG was increased (*p* < 0.05). CUMS induced-decreases in the NE and 5-HT concentrations were respectively restored by EPA (*p* < 0.05) and DHA (*p* < 0.05) supplementation, whereas the decreased NA/MHPG ratio induced by CUMS was equally attenuated by EPA or DHA addition (Table 2). Compared to control group, although lower ratios of 5-HT/5-HIAA, and DA/DOPAC were seen in the CUMS group, the results were not statistically significant.

### 2.5. Cytokine Concentrations in the Hippocampus

As shown in Figure 3, compared to the control group, concentrations of IL-1β, IL-6 and TNF-α in the hippocampus were significantly increased after CUMS administration (*p* < 0.05 respectively). EPA supplementation significantly attenuated the increased IL-1β, IL-6 and TNF-α contents in the hippocampus, while DHA diet only decreased the IL-1β level (*p* < 0.05) (Figure 3A–C).

### 2.6. Changes in Glial Activation

Compared to the control group, microglia marker CD11b expression in the hippocampus was increased after CUMS exposure (*p* < 0.05), indicating microglia cells were activated by CUMS. However, increased CD11b expression was significantly attenuated by either EPA or DHA supplementation (*p* < 0.01, respectively). By contrast, decreased hippocampal GFAP (astrocyte marker) expression was found in CUMS rats (*p* < 0.01), indicating that the activity of astrocytes in the hippocampus was inhibited by CUMS. It is worth noting that decreased GFAP expression was only restored by EPA treatment (*p* < 0.01) (Figure 4A).

### 2.7. Hippocampal Expressions of BDNF, Trk B and p75NTR Receptors

The Western blotting analysis shows that hippocampal expression of BDNF and TrkB was decreased after CUMS exposure (*p* < 0.01, respectively), which were only reversed by EPA supplement, but not by DHA. Meanwhile, the expression of p75NTR in the hippocampus was increased followed CUMS exposure (*p* < 0.05), which could be attenuated by either EPA or DHA diet (*p* < 0.01 respectively) (Figure 4B).

### 2.8. Glial Cell-Derived Neurotrophic Factor (GDNF) and Nerve Growth Factor (NGF)Expressed in the Hippocampus

The expression of GDNF in the hippocampus was significantly decreased after CUMS exposure (*p* < 0.01), which was significantly reversed by intakes of either EPA or DHA. However, the effect of EPA on GDNF expression was more powerful than that of DHA (EPA; *p* < 0.01, DHA; *p* < 0.05). A similar pattern was found at the expression of NGF. Reduced NGF expression was observed in rats exposed to CUMS (*p* < 0.01), which could be rescued by the consumption of either EPA or DHA (*p* < 0.01, *p* < 0.05 respectively). Again, the ability of EPA to attenuate decreased NGF expression was much stronger than that of DHA (*p* < 0.01) (Figure 5A).

### 2.9. The Activity of NF-ΚB and p38 Pathways in the Hippocampus

As shown in the Figure 5, the expression of NF-ΚB and p38 in the hippocampus was increased by the CUMS (*p* < 0.01 respectively), both of which could be downregulated by intakes of either EPA or DHA (*p* < 0.01 respectively). However, the upregulation effect of EPA (*p* < 0.05) on down-regulating NF-ΚB expression was better than that of DHA (*p* < 0.05), whereas no difference between the effect of EPA and DHA on p38 activity was found (Figure 5B).

### 2.10. Changes of Apoptosis Related Factors in the Hippocampus

Compared to the control group, the hippocampal expression of bax was increased in the CUMS group (*p* < 0.05), which could be dramatically decreased by consumption of either EPA or DHA (*p* < 0.01 respectively) (Figure 6A,B). Again, the effects of EPA on modulating bax activity was more powerful than that of DHA. Conversely, the content of bcl-2 in the hippocampus was significantly declined under the stressed circumstance (*p* < 0.01), which could be elevated by the intake of EPA (*p* < 0.01), whereas no effect was found by DHA addition (Figure 6A,B). The ratio of bax: bcl-2 was increased by the CUMS administration (*p* < 0.01), which could be reduced by both EPA and DHA in the diet (EPA: *p* < 0.01, DHA: *p* < 0.05 respectively) (Figure 6A,B). Similarly, the effect of EPA on moderating the ratio of bax: bcl-2 was better than that of DHA (*p* < 0.05). (Figure 6A,B)

## 3. Discussion

The present study demonstrated that CUMS administration could induce neuroinflammation and neurotrophins deficits in the hippocampus [6], as showed increased pro-inflammatory cytokines (IL-1β, IL-6, TNF-α), and deceased expression of BDNF, GDNF and NGF. CUMS exposure also stimulated the HPA axis to release more corticosterone and caused 5-HT and NE deficit in the hippocampus. Furthermore, imbalanced n-3 and n-6 PUFA profile in the brain and elevated total cholesterols in the serum were also observed in the CUMS rats. All these changes were in line with those observed in depressed patients.

Previous clinic studies suggested that purified EPA or EPA-enriched, rather than purified DHA or DHA enriched supplements, are more effective in the prevention and treatment of depression [28,29,30,31,32]. The behavioral results from the present study partially support the clinical findings. EPA but not DHA dietary exhibited effect in the OFT. Even though anhedonia behavior in the SPT, and depressive-like behavior in the FST were ameliorated by both supplementation, EPA effects on these behavioral abnormalities were much more powerful than DHA. Then, the study for the first demonstrated that the mechanisms by which EPA or DHA attenuated CUMS-induced depression-like changes. The possible difference between these two PUFAs are discussed below.

First, in line with clinic results [19,20], this study demonstrated that CUMS induced depression-like behaviors were associated with higher n-6 AA and lower n-3 PUFAs (EPA, DPA and DHA) levels in the brain. Although higher AA content induced by CUMS could be reduced via either EPA or DHA intake, lower DPA and EPA levels were only elevated by EPA supplementation. Contrarily to our results that brain levels of EPA and DHA were not affected by DHA addiction, some researchers reported that an increase in DHA consumption can result in an increase of EPA levels in plasma [33]. However, evidence from both rats and human indicates that the increase in EPA is more likely because of a slower turnover of EPA in the plasma and liver, rather than the retro-conversion pathway of DHA to EPA [33,34]. Nevertheless, DHA feeding may slow the conversion of EPA to downstream metabolic products in the liver, with a concomitant decrease of DPA levels seen after DHA supplementation [33,34,35]. Taken together, the current data may suggest that the antidepressant-like effects of EPA better than DHA were possibly not through DHA, but rather due to EPA own function, either directly or via its metabolite DPA. Since the function of DPA in the brain is less understood, this possibility should be further investigated.

Second, stress-induced dyslipidemia is also commonly observed in patients with anxiety or depressive disorders and in animal models [6,36,37]. As essential components of cell membranes, EPA and DHA interact with other cell membrane components, increase membrane fluidity, eventually affecting the activity of many membrane-bound proteins [29,38]. However, increased membrane cholesterol could reduce membrane fluidity [39]. The present study demonstrates that EPA diets were more effective than DHA to lower total cholesterol concentration in serum. Due to laboratory limitations, we did not measure membrane fluidity of neuron cells in the brain; thus, the exact effect of EPA and DHA regulating membrane fluidity remains unknown. However, the present study not only demonstrated the correlation between n-3 PUFA attenuation of depression and CUMS-induced cholesterol changes, but also distinguished EPA effect from DHA on stress-increased serum cholesterol.

Third, consist with previous studies [40,41], the present study shows that CUMS induced-depression-like behaviors were associated with decreased NE and 5-HT contents in the hippocampus. Then, this study, for the first, shows that the decreased NE was only increased by EPA, while reduced 5-HT was only elevated by DHA. Since NE in the brain mainly contribute to reward behavior and positive emotion in depression [42,43], our results show that EPA was better effects than DHA on anhedonia (more sucrose consumption in SPT) and reverse desperate state in CUMS rats (shorter immobility time in FST). Conversely, the 5-HT deficit in depression was associated with negative emotions, such as anxiety, fear, hostility, and irritability [5]. Considering the traditional behavioral experiment is mainly focused on abatement of positive emotion, whether the anxiety behavior of rats increased or not should be investigated in future work.

Fourth, as mentioned previously, inflammation may play an important role in the etiology of depression [44]. The present study, for the first time, reported more profound anti-inflammatory effects of EPA than DHA on microglia M1-induced neuroinflammation, such as EPA decreasing IL-1β, IL-6, TNF-α, but DHA only reducing IL-1β levels in the hippocampus of depression model, which should be taken into account for a better clinical outcome. An important finding directly support the point is that EPA contents in microglial cells appears to be at least two-fold higher than DHA [33]. Although the mechanism by which EPA is better than DHA to modulate microglia activity remains unclear, the present study shows that EPA was better to suppress neuroinflammatory pathways, such as stronger inhibiting NF-κB expression in the hippocampus. Taken together, these results all support the stronger anti-inflammatory effects of EPA than DHA in the depression model.

Fifth, we previously demonstrated that microglial activation may suppress astrocyte functions, such as reducing the expression of neurotrophins [6,45]. The present study for the first time found that EPA supplementation was better than DHA to reverse decreased astrocyte activities, with the evidences of better increasing BDNF, NGF and GDNF expression in the hippocampus. Binding to TrkB receptors, BDNF activates several intracellular pathways, including mitogen-activated protein kinase (MAPK) and phosphoinositide 3-kinase (PI3 K)/Akt/Bcl2, which promote neurogenesis, neuronal differentiation and survival [46]. However, TrkB reduction can trigger p75NTR receptor expression, thereby inducing neuron apoptosis. The present study, on the one hand, demonstrated that EPA is better than DHA to up-regulate the decreased expression of astrocyte marker GFAP, as well as BDNF and TrkB expression induced by CUMS. On the other hand, EPA was stronger than DHA to inhibit apoptosis-related bax expression and reverse the lower bcl-2 expression caused by CUMS administration. Thus, EPA may exert better neuroprotective functions than DHA by modulating BDNF, TrkB receptors, and the bcl-2 pathway.

## 4. Materials and Methods

### 4.1. Animals and Experimental Procedure

Forty adult female Sprague Dawley (SD) rats (8–12 weeks, 280–300 g) were obtained from the Southern Medical University (Guangzhou, China), and production license No. SCXK (Yue) 20110015. Rats were housed as 2 per cage under standard conditions (12-h light/dark cycle, 23 ± 2 °C and relative humidity of 50–60%, free access to food and water). The experimental protocol was approved by the Animal Care and Use Committee of Guangdong Ocean University, China (IACUC-20160922-037), and conducted strictly in compliance with the Guide for Care and Use of laboratory animals (Chinese Council). The experiments were conducted in a blinded fashion

Prior to the experimental procedure, rats were adopted the laboratory environment for at least 1 week. Then, rats were exposed to two different mild, unpredictable stressors every day (with no stressor repeated within 3 days) as our previous study for 45 days [6], and fed a diet according to its grouping. After depression-like behaviors were evaluated, animals were decapitated. Blood samples were collected, and then brain tissues were carefully dissected on dry ice and frozen in liquid nitrogen rapidly, which were stored in −80 °C until further measurements (Figure 7).

### 4.2. Diets

Animals were divided into four groups (*n* = 10) as follows: CT (palm oil supplemented diet); CUMS (CUMS+ palm oil supplemented diet); CUMS + EPA (CUMS + EPA supplemented diet); CUMS + DHA (CUMS + DHA supplemented diet). Rats were fed a diet consisting of regular chow powder (Guangdong Medical Laboratory Animal Center, Zhanjiang, China) mixed with 1% palm oil (GuoYang Biotech Company, Guangzhou, China) or 1% ethyl-EPA (96% pure) (Renpu pharmaceutical co. LTD, Suzhou, China) or 1% DHA (96% pure) (Renpu pharmaceutical co. LTD, Suzhou, China), as described previously [22,47]. Palm oil was chosen as the control diet since it contains low amounts of n-6 fatty acids and negligible amounts of n-3 fatty acids, and has comparable caloric values [21]. Diets for each group were prepared according to the previous method with slight modification [21]. Briefly, the master bottles of palm, EPA, and DHA oil were taken from −20 °C freezer, carefully mixed with regular chow powder, then aliquoted to Falcon tubes that were flushed with nitrogen gas to prevent oxidation. The diets were prepared every 2 days and flushed with nitrogen, sealed and stored in 4 °C to prevent oxidation. The diet was fed before the light turning off and last for 12 h, and the leftover was then taken away every day.

### 4.3. Behavior Tests

#### 4.3.1. Sucrose Preference Test

SPT was used to measure the anhedonia state of rats [48]. After CUMS administration, two-bottle of 1% sucrose solution were given to all animals at the first day of SPT. One bottle of 1% sucrose was changed with pure water on following day. Then, all animals were deprived of any drinking solution and food for 24 h. Subsequently, each cage was provided two pre-weighed bottles solution (one containing 1% sugar water and another one containing regular drinking water), and rats were free access to solutions and food for 1 h. Then, two bottles weighed to calculate sucrose preference (sucrose solution intake over total drinking solution intake).

#### 4.3.2. Forced Swimming Test

FST was performed for assessing giving up-like behavior and test [49]. Briefly, rats were forced to swim adaptively 10 min in the container one day before the test. Then, rats were individually placed in an inescapable transparent cylinder (20 cm in diameter, 50 cm in height) containing water (25 °C). The immobile time as ceased struggling and remained floating motion in the water was recorded blindly by Super Maze Behavior Analysis System (Shanghai Xinruan Information Technology Co., Ltd., Shanghai, China) for 5 min. Water in the cylinder was changed after each test, and rats were dried with towels and then transferred to their dry cages.

#### 4.3.3. Open Field Test

OFT was used to evaluate the spontaneous locomotor activity and exploratory behaviors in a novel environment [50]. It was assessed in a round open field apparatus (diameter 100 cm, high 50 cm) with white painted wall and floor. A 60 W white bulb was positioned 100 cm above the center of the apparatus [25]. Rats were gently placed in the central of apparatus with head facing toward the wall to observe the behavior for 3 min. Then, the number of locomotor crossing (the number of squares crossed), rearing times (when a rat stood completely erect on its hind legs) were recorded using the Super Maze behavior analysis system (Shanghai Xinruan Information Technology Co., Ltd., Shanghai, China) by two highly trained observes blinded to the treatment groups. At the end of each test, the apparatus was cleaned with a wet towel.

### 4.4. Lipid Extraction and Gas Chromatography (GC)Analysis for Brain n3/n6 PUFA Profile

The n-3 and n-6 PUFA profile in the brain were evaluated by GC according to the previous method [51,52]. Brain tissues were homogenized with chloroform/methanol solution (2/1), and centrifuged with 0.9% NaCl solution at 500 rap/min for 20 min. The bottom phase containing lipids was collected and dried with nitrogen stream. Then, dried fatty acid samples were incubated with 1.5% sulfuric acid methanol and methylene chloride solution in a water bath at 100 °C for 1 h. After cooling, hexane and H2O were added and centrifuged at 500 rpm for 2 min. The hexane phase containing fatty acid methyl ester (FAME) lipids was dried by nitrogen and collected for GC analysis. FAME was analyzed by GC using an Agelent gas chromatograph (CA, USA) 1260 equipped with a flame ionization detector. The FAME was separated on a 60 m Agilent 122- 2361 DB-23 capillary column (0.25 mm diameter and 0.15 μm coating thickness) using helium at a flow rate of 1.8 mL/min. The inlet and detector temperatures were constant at 260 °C. By comparing retention times with external FAME standard mixtures (Nu-Chek-Prep, Main St. Elysian, MN, USA), and docosapentaenoic acid (DPA) PUFA standard (Nu-Chek-Prep, U-101-M, Main St. Elysian, MN, USA), n-3 and n-6 PUFA contents in the brain were identified and measured.

### 4.5. Measurement of Serum Total Cholesterol and Corticosterone Levels

Whole blood of rats was collected and stand at room temperature for 2 h. Then, serum samples were taken after centrifugation at 1400 r/m at room temperature for 10 min and stored at −80 °C until assayed. The levels of serum total cholesterol and corticosterone were measured by commercial kits from Shanghai Enzyme-linked Biotechnology Co., Ltd. (Shanghai, CN) in accordance with the manufacturer′s protocols.

### 4.6. High-Performance Liquid Chromatography (HPLC) Analysis of Neurotransmitters and Metabolites in the Hippocampus

Hippocampal tissues were homogenized in 0. 60 M ice-cold perchloric acid containing 50 mM Na_2_EDTA with 100 ng isoproterenol as an internal standard. The homogenates were centrifuged twice at 14,000 rpm/min for 15 min at 4 °C. Supernatants were filtrated with 0.45 µm membrane and transferred to new tubes. The PH of supernatants was adjusted to 3.8 with 1 M sodium acetate, then supernatants were stored at −80 °C until use. For the HPLC analysis, 10 µL of the pH-adjusted supernatant was injected into an HPLC system with fluorescence detection (Agelent, Santa Clara, CA, USA). The neurotransmitters and their metabolites in samples was separated by a C18 reverse-phase column (4.5 × 150 mm) (Agelent, Santa Clara, CA, USA) with a mobile phase containing sodium acetate and citric acid. The mobile phase was prepared as follow: 0.1 M sodium acetate was mixed 0.1M citric acid in a 10:9 ratio, and adjusted to PH 3.5 (0.1 M sodium citric buffer), mixed with methanol in a ratio of 85:15 and then supplemented with sodium octane sulfonate (100mg/L), Na_2_EDTA (5mg/mL). Dopamine (DA), 3,4dihydroxyphenylacetic acid (DOPAC), homovanillic acid (HVA), NE, 3-methoxy-4-hydrophenyl (MHPG), 5-HT, and 5-hydroxyindole-3-aceticacid(5-HIAA) were analyzed.

### 4.7. Measurement of Pro-Inflammatory Cytokines Contents in the Hippocampus by ELISA Kits

The concentrations of IL-1β, IL-6, and TNF-α in the hippocampus were measured by commercial ELISA kits from Shanghai Enzyme-linked Biotechnology Co., Ltd. (Shanghai, CN) according to the manufacturer′s protocol.

### 4.8. Western Blotting

According to its manufacturer′s instructions, the total protein of hippocampus was extracted with a commercial kit (Beyotime technology, China). The concentration of the protein was measured with BCA kit (Beyotime technology, China). Subsequently, 30–50 μg protein was loaded onto a 10% polyacrylamide gel, and then transferred to PVDF (Millipore Corp., Burlington, MA, USA) membrane. After incubation with 5% non-fat milk at room temperature for one hour, membranes and incubated with primary antibodies at different dilutions overnight at 4 °C. After washing, the membranes were further incubated with special secondary antibody, followed by detection using enhanced chemiluminescence (Millipore Corp., Burlington, MA, USA). The bands were scanned and analyzed using a chemiluminescence system (Tanon 5200, Shanghai, China). The mouse monoclonal primary antibodies for anti-GDNF (sc-13147,15kDa, 1:1000), anti-NGF (sc-365944, 13 kDa, 1:1000), anti-Bax (sc-7480, 23 kDa, 1:1000), anti-Bcl-2 (sc-7382,26 kDa, 1:1000), anti-NF-Κb (sc-8008,65 kDa, 1:1000), anti-P38 (sc-81621,38 kDa, 1:1000), GFAP (sc33673, 50 kDa, 1:500) and β-Actin (sc-47778, 43 kDa) were purchased from Santa Cruz Biotechnology, USA, while the rabbit monoclonal primary antibodies for anti-CD11b (ab75476, 160 kDa, 1:800), BDNF (ab108319, 15 kDa, 1:2000), p75 (ab52987, 75 kDa, 1:2000) and Trk B (ab18987, 92 kDa, 1:1500) were purchased from Abcam Trading Company Ltd. (Shanghai, China).

### 4.9. Statistical Analysis

Except for 6 independent experiments were performed for protein expressions by western blot, the other measurements were in 10 animals. The results are expressed as mean ± SEM and analyzed by SPSS version 17. Statistical analyses were performed by one-way analysis of variance (ANOVA) followed by TUKEY post hoc as multiple comparisons between groups. Significance was indicated at a *p*-value < 0.05 and *p*-value < 0.01.

## 5. Conclusions

Taken together, by using the CUMS depression model, the present study for the first demonstrated that EPA is more effective than DHA to attenuate CUMS-induced depression-like behaviors. A possible mechanism may be associated with (1) EPA is better than DHA to modulate CUMS-induced n-3/n6 PUFA imbalance profiles and higher serum total cholesterol level; (2) EPA is better than DHA to inhibit microglial activation and associated neuro-inflammation and NF-kB signaling; (3) EPA is better than DHA to improve decreased astrocytes activity and promote GDNF, NGF, BDNF neurotrophins expression; (4) EPA is better than DHA to protect neuro cells form death by bcl-2 family and BDNF-TrkB signaling in the hippocampus. A schematic explanation for how CUMS inducing depressive-like behaviors and the role of EPA and DHA is shown in Figure 8.

## Figures and Tables

**Figure 1 ijms-21-01769-f001:**
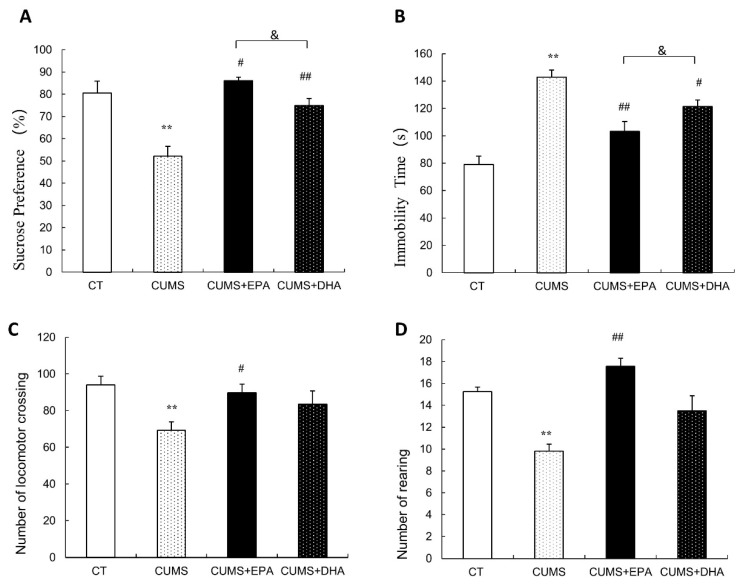
Behavioral tests. (**A**) Percentage of sucrose consumption in the sucrose preference test. (**B**) Immobility time in the forced swimming test. (**C**) Number of crossing in the open field test. (**D**) Number of rearing in the open field test. ** *p* < 0.01, CUMS vs. CT; ^#^
*p* < 0.05, ^##^
*p* < 0.01, CUMS +EPA vs CUMS, CUMS +DHA vs CUMS; ^&^
*p* < 0.05, CUMS + EPA vs. CUMS +DHA.

**Figure 2 ijms-21-01769-f002:**
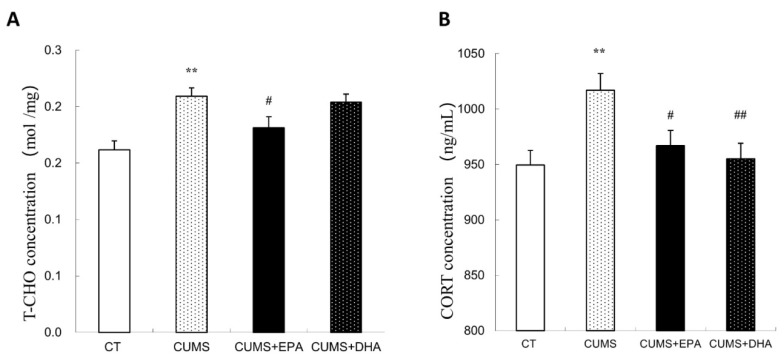
The concentration of total cholesterol (T-CHO) (**A**) and corticosterone (CORT) (**B**) in the serum. Data are expressed as mean ± SEM. ** *p* < 0.01, CUMS vs CT; ^#^
*p* < 0.05, ^##^
*p* < 0.01, CUMS + EPA vs CUMS, CUMS + DHA vs CUMS.

**Figure 3 ijms-21-01769-f003:**
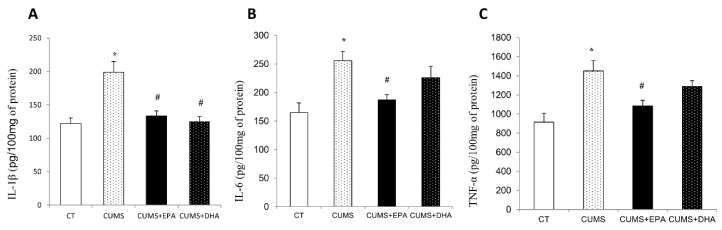
The concentrations of pro-inflammatory cytokines in the hippocampus. (**A**) IL-1β concentration in the hippocampus. (**B**) IL-6 concentration in the hippocampus. (**C**) TNF-α concentration in the hippocampus. data was expressed as mean ± SEM. * *p* < 0.05, CUMS vs CT; ^#^
*p* < 0.05, CUMS + EPA vs CUMS, CUMS + DHA vs CUMS.

**Figure 4 ijms-21-01769-f004:**
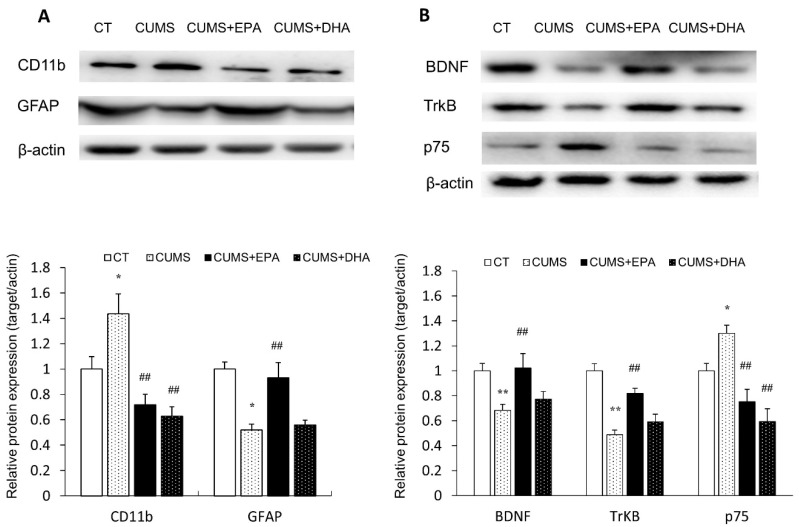
Protein expressions of CD11B, GFAP, BDNF, TrkB, p75, in the hippocampus of rats. (**A**) The protein expressions of microglial marker CD11b, astrocyte marker GFAP in the hippocampus of rats; (**B**) Protein expressions of BDNF and its receptor Trkb and p75 in the hippocampus of rats. Data were normalized to protein expression of housekeeping gene beta-actin using the delta-Ct method, and expressed as mean ± S.E.M fold of control.* *p* < 0.05, ** *p* < 0.01,CUMS vs CT; ^##^
*p* < 0.01, CUMS +EPA vs CUMS, CUMS +DHA vs CUMS.

**Figure 5 ijms-21-01769-f005:**
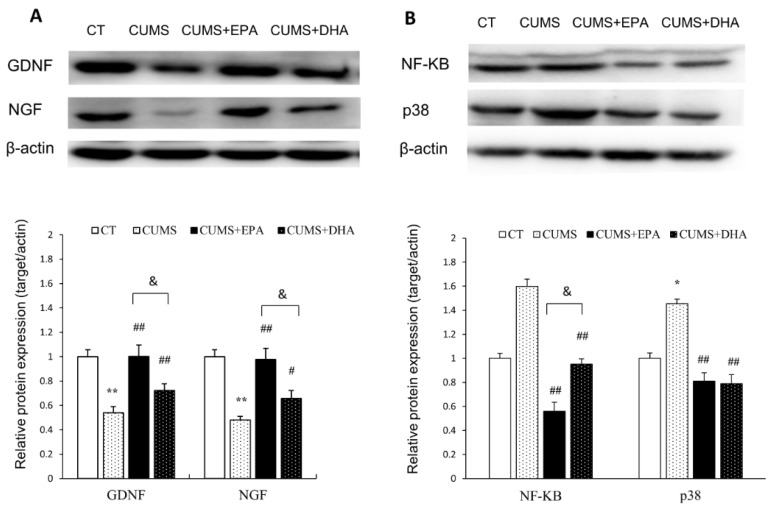
Protein expressions of GDNF, NGF, NF-KB, and p38 in the hippocampus of rats. (**A**) GDNF and NGF expression in the hippocampus of rats; (**B**) of NF-ΚB and p38 expression in the hippocampus of rats. Data were normalized to protein expression of housekeeping gene beta-actin using the delta-Ct method, and expressed as mean ± S.E.M fold of control.* *p* < 0.05, ** *p* < 0.01, CUMS vs CT; ^#^
*p* < 0.05, ^##^
*p* < 0.01,CUMS +EPA vs CUMS, CUMS +DHA vs CUMS; ^&^
*p* < 0.05,CUMS +EPA vs CUMS +DHA.

**Figure 6 ijms-21-01769-f006:**
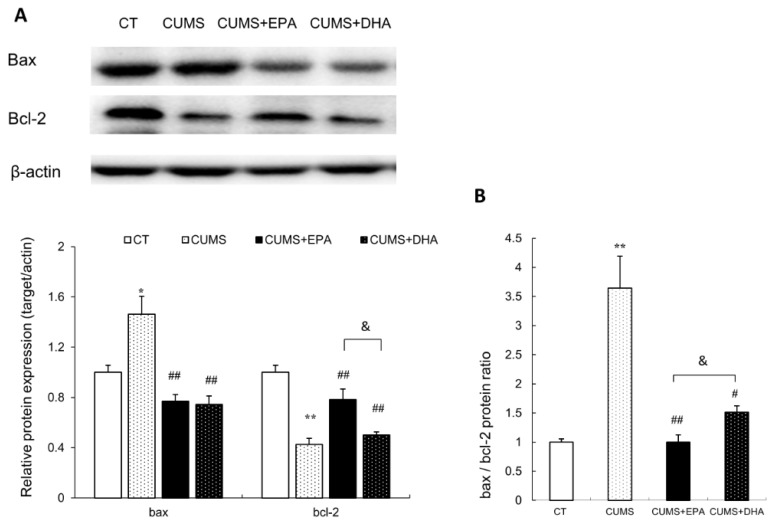
Expression of apoptosis related factors in the hippocampus of rats. (**A**) Protein expressions of bax and bcl-2 in the hippocampus of rats, (**B**) bax/bcl-2 ratio in the hippocampus of rats. (**A**) Data were normalized to protein expression of housekeeping gene beta-actin using the delta-Ct method, and expressed as mean ± S.E.M fold of control. * *p* < 0.05, ** *p* < 0.01, CUMS vs CT; ^#^
*p* < 0.05, ^##^
*p* < 0.01, CUMS +EPA vs CUMS, CUMS + DHA vs CUMS; ^&^
*p* < 0.05, CUMS +EPA vs CUMS +DHA.

**Figure 7 ijms-21-01769-f007:**
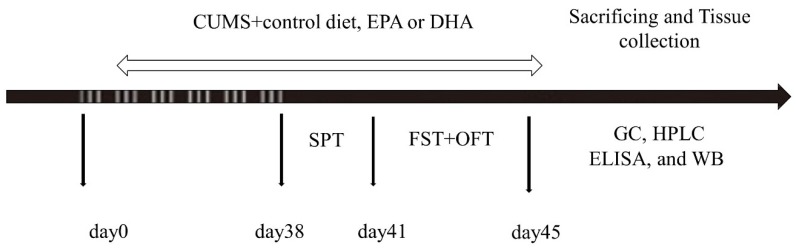
Schedule of the experimental procedures. CUMS: chronic unpredictable mild stress; SPT: sucrose preference test; FST: forced swimming test; OPT: open field test; GC: gas chromatography; HPLC: high performance liquid chromatography; ELISA: Enzyme-linked immunosorbent assay; WB: Western blotting.

**Figure 8 ijms-21-01769-f008:**
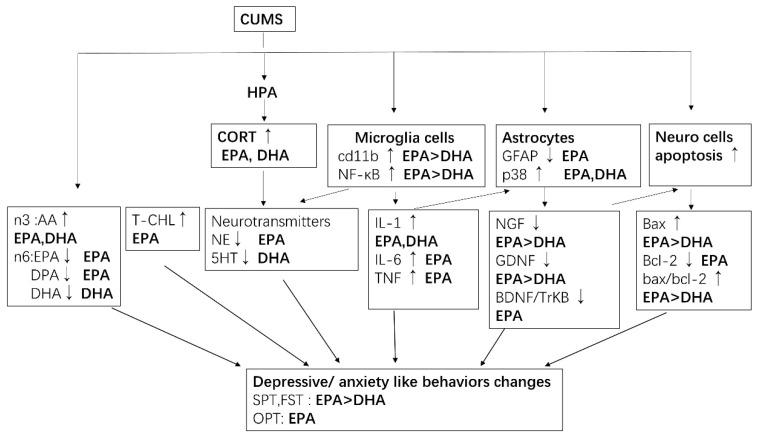
Possible mechanism by which EPA and DHA attenuated CUMS-induced depression-like changes. ↑: level was increased, ↓level was decreased. Deep black color represents effective.

**Table 1 ijms-21-01769-t001:** Fatty Acid profile in rats brain tissue.

Fatty Acid mg/g Tissue	CT	CUMS	CUMS + EPA	CUMS + DHA
LA (18:2 n-6)	0.326 ± 0.050	0.362 ± 0.046	0.364 ± 0.043	0.428 ± 0.047
GLA (18:3 n-6)	0.348 ± 0.034	0.297 ± 0.034	0.400 ± 0.069	0.434 ± 0.075
DGLA (20:3 n-6)	0.325 ± 0.065	0.355 ± 0.036	0.404 ± 0.066	0.407 ± 0.060
AA (20:4 n-6)	1.189 ± 0.082	1.661 ± 0.113 *	1.241 ± 0.132 ^#^	1.141 ± 0.126 ^##^
ALA (18:3, n-3)	0.504 ± 0.089	0.379 ± 0.070	0.412 ± 0.072	0.538 ± 0.133
EPA (20:5 n-3)	0.230 ± 0.032	0.120 ± 0.027 **	0.591 ± 0.057 ^##^	0.164 ± 0.038
DPA (22:5 n-3)	0.346 ± 0.029	0.155 ± 0.015 *	0.481 ± 0.170 ^##^	0.232 ± 0.043
DHA (22:6 n-3)	2.374 ± 0.177	1.596 ± 0.182 *	1.908 ± 0.107	2.650 ± 0.303 ^##^

LA: linolcic acid; GLA: γ-linolenic acid; DGLA: dinomo-γ-linolenic acid; AA: arachidonic acid; ALA: α-Linolenic acid; EPA: eicosapentaenoic acid; DPA: docosapentaenoic acid; DHA: docosahexaenoic acid. Data are expressed as mean ± SEM. * *p* < 0.05; ** *p* < 0.01, CUMS vs CT; ^#^
*p* < 0.05, ^##^
*p* < 0.01, CUMS +EPA vs CUMS, CUMS +DHA vs CUMS.

**Table 2 ijms-21-01769-t002:** Neurotransmitters and their metabolites concentrations in the hippocampus.

Neurotransmitters ng/mg Tissue	CT	CUMS	CUMS + EPA	CUMS + DHA
MHPG	9.258 ± 0.223	10.239 ± 0.264 *	9.750 ± 0.274	9.697 ± 0.312
NE	15.968 ± 0.474	13.984 ± 0.202 *	16.462 ± 0.523 ^##^	15.364 ± 0.833
DA	3.045 ± 0.072	3.086 ± 0.086	3.048 ± 0.085	3.156 ± 0.125
DOPAC	0.090 ± 0.011	0.094 ± 0.010	0.108 ± 010	0.119 ± 0.017
5-HT	1.089 ± 0.027	0.960 ± 0.027 **	1.014 ± 0.024	1.090 ± 0.012 ^##^
5-HIAA	0.263 ± 0.015	0.265 ± 0.012	0.278 ± 0.020	0.271 ± 0.019
HVA	0.477 ± 0.115	0.331 ± 0.046	0.566 ± 0.183	0.749 ± 0.314
NE/MHPG	1.607 ± 0.111	1.330 ± 0.071 *	1.610 ± 0.060 ^#^	1.66 ± 0.27 ^#^
5-HT/5-HIAA	4.123 ± 0.266	3.997 ± 0.257	3.743 ± 0.229	3.932 ± 0.169
DA/DOPAC	39.107 ± 8.438	34.868 ± 4.268	29.616 ± 3.191	28.873 ± 3.391
DA/HVA	8.488 ± 1.933	10.411 ± 1.640	7.538 ± 1.375	6.729 ± 1.317

MHPG:3-methoxy-4-hydrophenyl, NE: noradrenaline, DA: Dopamine, DOPAC:3,4dihydroxyphenylacetic acid, HVA: homovanillic acid, 5-HT: serotonin, 5-HIAA: 5-hydroxyindole-3-aceticacid. Data are expressed as mean ± SEM. * *p* < 0.05; ** *p* < 0.01, CUMS vs CT; ^#^
*p* < 0.05, ^##^
*p* < 0.01, CUMS +EPA vs CUMS, CUMS +DHA vs CUMS.

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
