# Peer review of "EPA is More Effective than DHA to Improve Depression-Like Behavior, Glia Cell Dysfunction and Hippcampal Apoptosis Signaling in a Chronic Stress-Induced Rat Model of Depression"

_ijms, 2020, doi:10.3390/ijms21051769_

Round 1

Reviewer 1 Report

This is a nice little study that describes the advantage of EPA over DHA to improve depression symptoms in rats that were exposed to chronic mild stress. 

i have a few questions before i think this study is ready for publication. the English needs some brushing up in places.

abbreviations should be explained when first used, e.g. ROS and in general, there are so many abbreviations used that it makes it extremely hard to read the manuscript.

figure 1: abbreviations should be explained in the figure caption  so the reader does not have to go back to the text

The discussion is giving a massive overview of what is known from the literature but the authors seem to not integrate that too well with their own findings, this can certainly be improved. maybe even the discussion could be straightened a bit as it seems very lengthy. even though the authors set out to answer compare and evaluate the different mechanisms of EPA and DHA in the treatment of depression, they still leave the reader with a lot of speculation.

Materials and methods: 4.1. the authors do need to provide the number from the local ethics committee that has granted the work presented; 4.2. what was the DHA and EPA dissolved in, has this been given as a vehicle to the control and CUMS only animals to exclude other effects than the PUFAs to play a role here.. what is the bioavailability of 1% ethyl EPA, how stable are EPA and DHA in the animal chow, how often was the chow replaced? 4.8. why was skin used to perform the western blot experiments?; 4.9. how many independent experiments were performed for the statistical analysis. why was a post hoc analysis performed?

Author Response

Thank you for giving us the opportunity to revise our manuscript. For your convenience, all revised items are now highlighted in red color in the resubmitted manuscript text file.

Point 1. This is a nice little study that describes the advantage of EPA over DHA to improve depression symptoms in rats that were exposed to chronic mild stress. I have a few questions before I think this study is ready for publication. the English needs some brushing up in places.

Reply: Thanks to Reviewer 1 for the positive evaluation. The language issue has been modified.

Point 2. Abbreviations should be explained when first used, e.g. ROS and in general, there are so many abbreviations used that it makes it extremely hard to read the manuscript.

Reply:  These have been corrected.

Point 3. Figure 1: abbreviations should be explained in the figure caption so the reader does not have to go back to the text.

Reply:  These have been explained as requested.

Point 4. The discussion is giving a massive overview of what is known from the literature, but the authors seem to not integrate that too well with their own findings, this can certainly be improved. maybe even the discussion could be straightened a bit as it seems very lengthy. even though the authors set out to answer compare and evaluate the different mechanisms of EPA and DHA in the treatment of depression, they still leave the reader with a lot of speculation.

Reply: The suggestion has been well accepted. In the revised manuscript, each finding from the present study has been discussed and then linked to supporting literatures.

Point 5. Materials and methods: 4.1. the authors do need to provide the number from the local ethics committee that has granted the work presented

Reply: We have now provided approval number by the institutional Bioethics Committee, as requested (lines 302)

Point 6. Materials and methods: 4.2. what was the DHA and EPA dissolved in, has this been given as a vehicle to the control and CUMS only animals to exclude other effects than the PUFAs to play a role here. what is the bioavailability of 1% ethyl EPA, how stable are EPA and DHA in the animal chow, how often was the chow replaced?

Reply: The chow powder was mixed with 1% palm oil, 1% ethyl EPA or 1% DHA. In order to prevent oxidation and keep the PUFAs stable, the master bottles of palm, EPA, and DHA oil were taken from −20 °C freezer, carefully mixed with regular chow powder under nitrogen gas, then aliquoted into Falcon tubes that were flushed with nitrogen gas. The diets were prepared every 2 days and sealed and stored in 4 °C to prevent oxidation. The diet was fed before the light turning off and last for 12 h. The leftover was then taken away every day. Those details were added into materials and methods of the revised manuscript, and our previous publication with this method has been also cited.

Point 7. Materials and methods: 4.8. why was skin used to perform the western blot experiments.

Reply: Sorry for the mistake. It has been corrected to “hippocampus”.

Point 8. Materials and methods: 4.9. how many independent experiments were performed for the statistical analysis; why was a post hoc analysis performed?

Reply: The number of each independent experiment was added to statistic section as requested. As for the statistical analysis, TUKRY post hoc was performed as multiple comparisons between groups followed one-way ANOVA. This has been added to the statistical section.

Reviewer 2 Report

I have read the paper entitled "EPA is more effective than DHA to improve depression-like behavior, glia cell dysfunction and hippcampal apoptosis signaling in a chronic stress-induced rat model of depression" by Zhi-Lan Peng and collaborators. The paper assess that  EPA is more powerful than DHA in anti-inflammation, normalizing astrocyte and neurotrophin function and regulating NF-κB, p38 and apoptosis signaling. 

The paper is interesting and well conducted.

This reviewer has only a few concerns:

1) Authors should explain the reason why they use only female rats.

2) The quality of the representative western blots and the graph images is scarce. This has to be improved.

Author Response

Response to Reviewer 2 Comments

Thank you for giving us the opportunity to revise our manuscript. For your convenience, all revised items are now highlighted in red color in the resubmitted manuscript text file.

Point 1. The paper is interesting and well conducted. This reviewer has only a few concerns:1) Authors should explain the reason why they use only female rats.

Reply: Thanks to Reviewer 2 for the positive evaluation. In depressed patients, females are 2–3 times more likely than males. However, most scientists studied male animals in depression area for convenient references and stable results. We believe that female rat model of depression should be used and more appropriate in the any depression studies.

Point 2. The quality of the representative western blots and the graph images is scarce. This has to be improved.

Reply: In the revised manuscript, the figures has now been modified accordingly.

Round 2

Reviewer 1 Report

The authors have worked on the manuscript a good bit and improved it a lot by putting in additional information and clarification as requested. I am happy to see that the discussion flows so much better now.